

# Genetic insights into family group co-occurrence in *Cryptocercus punctulatus*, a sub-social woodroach from the southern Appalachian Mountains

Ryan C. Garrick

Department of Biology, University of Mississippi, Oxford, MS, United States of America

Corresponding author
Ryan C. Garrick,
rgarrick@olemiss.edu

## ABSTRACT

The wood-feeding cockroach *Cryptocercus punctulatus* Scudder (Blattodea: Cryptocercidae) is an important member of the dead wood (saproxylic) community in montane forests of the southeastern United States. However, its population biology remains poorly understood. Here, aspects of family group co-occurrence were characterized to provide basic information that can be extended by studies on the evolution and maintenance of sub-sociality. Broad sampling across the species' range was coupled with molecular data (mitochondrial DNA (mtDNA) sequences). The primary questions were: (1) what proportion of rotting logs contain two or more different mtDNA haplotypes and how often can this be attributed to multiple families inhabiting the same log, (2) are multi-family logs spatially clustered, and (3) what levels of genetic differentiation among haplotypes exist within a log, and how genetically similar are matrilines of co-occurring family groups? Multi-family logs were identified on the premise that three different mtDNA haplotypes, or two different haplotypes among adult females, is inconsistent with a single family group founded by one male–female pair. Results showed that of the 88 rotting logs from which multiple adult *C. punctulatus* were sampled, 41 logs (47%) contained two or more mtDNA haplotypes, and at least 19 of these logs (22% overall) were inferred to be inhabited by multiple families. There was no strong evidence for spatial clustering of the latter class of logs. The frequency distribution of nucleotide differences between co-occurring haplotypes was strongly right-skewed, such that most haplotypes were only one or two mutations apart, but more substantial divergences (up to 18 mutations, or 1.6% uncorrected sequence divergence) do occasionally occur within logs. This work represents the first explicit investigation of family group co-occurrence in *C. punctulatus*, providing a valuable baseline for follow-up studies.

## INTRODUCTION

In sub-social invertebrates, offspring stay with their parents for extended periods of time, but usually disperse before reproducing themselves (*Yip & Rayor, 2014*). Numerous studies—particularly those focusing on spiders—have sought to understand how this form

of social organization impacts genetic structure within species, and how cooperative group living evolves (e.g., *Johannesen et al., 1998*; *Johannesen & Lubin, 1999*; *Duncan et al., 2010*; *Yip, Rowell & Rayor, 2012*). The deepest insights into costs and benefits associated with transitions from sub-sociality to eusociality have been gained by studying closely related lineages that represent different stages along this gradient (*Bilde et al., 2005*; *Helantera et al., 2013*). Accordingly, identifying sets of taxa that are suitable for comparative analyses, and characterizing their basic population biology, is of considerable value.

Sub-social *Cryptocercus* woodroaches are the closest living relatives of extant termites (*Lo et al., 2000*). As a consequence of the phylogenetic position of *Cryptocercus* within Blattodea, members of this genus are key evolutionary links for understanding transitions to eusociality (*Klass, Nalepa & Lo, 2008*; *Nalepa, 2015*). The best studied members of *Cryptocercus* are the southern Appalachian Mountain lineages (i.e., the *C. punctulatus* complex) from the southeastern United States (*Bell, Roth & Nalepa, 2007* and references therein). Yet owing to their cryptic log-dwelling (saproxylic) nature, little is known about family group formation and co-occurrence in *C. punctulatus*, as direct observation is not possible in most cases. To extend our understanding of the ecological and microevolutionary processes that affect genetic structure and local persistence of sub-social invertebrates, these basic knowledge gaps need to be filled.

Previous work has revealed that *C. punctulatus* form mate pairs upon reaching maturity, and produce their first—and usually only—clutch of offspring approximately one year later. In addition to excavating galleries and maintaining the nest, parents provide intensive brood care. This includes feeding young instars on hindgut fluids via proctodeal trophallaxis, with care continuing after nymphs become nutritionally independent until the parents die, typically at least three years after the birth of their young (*Nalepa, 1984*; *Nalepa, 2015*). At this point, offspring are no longer fragile; they are at least half grown and have a robust cuticle (*Nalepa & Grayson, 2011*). Given that copulation between adults likely takes place within the same gallery of the rotting log in which the pair later raise their family, mate pairs have been considered monogamous since opportunities for extra-pair copulation are few (but see *Nalepa & Grayson, 2011*). Although the spatial demarcation of family groups is difficult, sampling strategies employed in several molecular studies of *C. punctulatus* have used a single random sample (e.g., *Steinmiller, Kambhampati & Brock, 2001*; *Aldrich, Krafsur & Kambhampati, 2004*; *Aldrich, Zolnerowich & Kambhampati, 2004*). Depending on the question at hand, this may be adequate. However, it is worth noting that large logs typically have numerous gallery systems (*Nalepa, 1984*) and so it is possible that multiple genetically divergent family groups do co-occur within a single log. But it is as-yet unknown if this occurs, how often, and from where multiple family groups tend to originate.

Recent work on *C. punctulatus* at Mountain Lake Biological Station in West Virginia has shed new light on dispersal and colonization processes, and mate-pair composition. A pitfall trapping study by *Nalepa & Grayson (2011)* confirmed that large nymphs, sub-adults, and adults do occasionally move between logs. Population genetic studies have repeatedly shown that dispersal distances of wingless saproxylic invertebrates are often very short (e.g., *Sunnucks et al., 2006*; *Garrick et al., 2007*; *Garrick et al., 2008*; *Leschen et al., 2008*; *Marske et al., 2009*; *Walker et al., 2009*; *Bull et al., 2013*), and this is also likely to be true of

*C. punctulatus* (*Nalepa et al., 2002*). Accordingly, following colonization of an uninhabited log by a male or female woodroach, potential mates probably arrive only from logs within close proximity. This suggests that *C. punctulatus* mate-pairs may be very close relatives. However, genetic estimates of relatedness between mate-pairs from 36 different logs at Mountain Lake showed that this is rarely the case (*Yaguchi et al., in press*). As a group, studies at this particular Biological Station (*Nalepa, 1984*; *Nalepa & Grayson, 2011*; *Yaguchi et al., in press*) have provided critical baseline data on the population biology *C. punctulatus*, but information about family group co-occurrence, and how this might vary across the species' range, is still lacking.

DNA sequence data from maternally-inherited mitochondrial DNA (mtDNA) have enabled inferences about the number and composition of family groups in diverse social animal species (*Möller et al., 2001*; *Faulkes, Arruda & Monteiro Da Cruz, 2003*), including insects (*Vargo, 2003*; *Holzer, Keller & Chapuisat, 2009*). The approach is most powerful when used in conjunction with information on the sex of sampled individuals. In the present study, range-wide geographic sampling of *C. punctulatus* was coupled with sexing and mtDNA sequencing of multiple adult cockroaches per log to address the following questions: (1) what proportion of rotting logs contain two or more different mtDNA haplotypes and how often can this be attributed to multiple families inhabiting the same log, (2) are multi-family logs spatially clustered, and (3) what levels of genetic differentiation among haplotypes exist within a log, and how genetically similar are matrilines of co-occurring family groups? Although these questions focus on basic characteristics of population biology, this work represents the first study to explicitly investigate family group co-occurrence in *C. punctulatus*.

## MATERIALS AND METHODS

### Taxonomy

The taxonomic status of southern Appalachian lineages of *Cryptocercus* is contentious. Briefly, following *Kambhampati, Luykx & Nalepa*'s (*1996*) discovery of four chromosomal races within the group, *Burnside, Smith & Kambhampati (1999)* described and named each of them as separate species. However, since no reliable morphological differences were apparent, the only diagnostic characters presented by the authors were mtDNA nucleotides (originally seven species-specific mutations, but subsequently reduced to four by *Steinmiller, Kambhampati & Brock, 2001*), and the descriptions were based on few reliably classified individuals (i.e., those for which both karyotype and mtDNA sequence were determined). Furthermore, species diagnosis on the basis of mtDNA has been applied inconsistently (e.g., *Aldrich, Krafsur & Kambhampati, 2004*), and the geographic origin of some type material is unclear (*Nalepa et al., 2002*). Since these issues remain unresolved, the original taxon name, *C. punctulatus*, is used here.

### Sampling and rotting log classification

Adult cockroaches ($n = 245$) were sampled from 88 rotting logs spanning the southern Appalachian Mountains and surrounding areas (Appendix S1), under scientific collecting permits issued by the Alabama DCNR, Georgia DNR (29-WBH-12-16), USDA Forest
**Table 1 Characteristics of mtDNA sequence data generated from 245 *C. punctulatus* individuals sampled in the present study.** Summary statistics are presented for each gene separately, and also for the concatenated dataset. Abbreviations are as follows: number of base pairs, bp; proportion of guanine plus cytosine nucleotides, G+C%; maximum-likelihood estimate (GTR+I+G model) of transition/transversion ratio, *ts/tv*; number of segregating sites, S; and number of unique haplotypes, $N_{hap}$.

| mtDNA gene region | Alignment length (bp) | G+C% | *ts/tv* | S | $N_{hap}$ | GenBank accessions |
|---|---|---|---|---|---|---|
| COI | 762 | 34.4 | 4.2 | 199 | 142 | KX944872–KX945114; and KU609620–KU609621 |
| COII | 363 | 25.7 | 6.7 | 109 | 64 | KX945115–KX945357; and KU609623–KU609624 |
| Concatenated | 1,125 | 33.8 | 4.9 | 308 | 155 | |

Service, and US National Park Service (GRSM-2012-SCI-2242; SHEN-2012-SCI-0015). Sampling involved breaking open the hard outer shell of each log with a hatchet and then carefully dismantling the interior woody material, which usually exhibited advanced-stage brown rot decay, with a small pry bar. Although it was not feasible to follow individual galleries, cockroaches were collected from the same general location within the log, until at least three adults had been caught. In most cases, broods were present with the sampled adults, but were usually not collected. Specimens will be lodged in the University of Mississippi Insect Collection (UMIC) following completion of an on-going project in which they are being used. Two or three adults per log (mean = 2.78) were used for molecular analyses. DNA extraction and polymerase chain reaction amplification, sequencing, alignment and validation of data from mtDNA *cytochrome oxidase subunit I* (COI) and *subunit II* (COII) genes followed *Garrick (2016)*, and characteristics of the molecular dataset are summarized in Table 1. For each individual, COI+COII sequences were concatenated (1,125-bp), and each rotting log was classified as containing cockroaches with the same mtDNA haplotype *vs.* two or more different mtDNA haplotypes (i.e., single-haplotype *vs.* multi-haplotype logs). For the latter group, the sex of each cockroach was determined based on presence (♂) *vs.* absence (♀) of styli on the ventral surface of the 9th abdominal segment (subgenital plate) via examination under 10× magnification. The null hypothesis of a 1:1 sex ratio was then assessed using a $\chi^2$ test.

The first goal of this study was to determine what proportion of rotting logs contain two or more different mtDNA haplotypes, and how often this can be attributed to multiple families inhabiting the same log. This information would provide insights into the basic population biology of *C. punctulatus*. To achieve this goal, individual-based information on mtDNA haplotype and sex (Appendix S2) was used to distinguish multi-haplotype logs that contained two or more different family groups (i.e., multi-family logs) from those that were consistent with expectations for only a single family (i.e., other multi-haplotype logs). The inference framework was based on the premise that mtDNA is strictly maternally inherited, and non-recombining. Furthermore, it was assumed that *de novo* mutations are sufficiently rare that their probability of occurrence within the 1,125-bp region sequenced in this study was zero. However, it was not assumed that all sampled adults from a given log were from the same age cohort. On this basis, a multi-family log was defined as any

Table 2 **All possible hypothetical combinations of mtDNA haplotype × individual sex, for a rotting log from which three adults were sampled.** For each row, different typeface (italics, underlined, or normal) represents different mtDNA haplotypes among individuals within a log, M indicates a male (i.e., father or son), and F indicates a female (i.e., mother or daughter). Multi-haplotype logs that unambiguously contain multiple families are labeled MF, whereas those which are consistent with the existence of only a single family are labeled OMH. Logs that contain a single mtDNA haplotype are designated SH.

| No. of different mtDNA haplotypes | Individual 1 | Individual 2 | Individual 3 | Rotting log classification |
| --- | --- | --- | --- | --- |
| 1 | M | M | M | SH |
| 2 | M | M | M | OMH |
| 3 | M | M | *M* | MF |
| 1 | F | F | F | SH |
| 2 | F | F | F | MF |
| 3 | F | F | *F* | MF |
| 2 | M | F | F | OMH |
| 2 | M | F | F | MF |
| 1 | M | F | F | SH |
| 3 | M | *F* | F | MF |
| 2 | M | M | F | OMH |
| 2 | M | M | F | MF |
| 1 | M | M | F | SH |
| 3 | M | *M* | F | MF |

log that contained three different haplotypes, or where two females each had a different haplotype (Table 2). All other situations can be attributed to the existence of only a single family derived from a monogamous pair of adults (e.g., via sampling a combination of mother, father, daughters and/or sons), and thus were designated as other multi-haplotype logs. These three log classes (Table 2) were the basis of subsequent analyses.

## Spatial clustering

The second goal of this study was to explore whether multi-family logs show a non-random spatial distribution; if so, this might indicate that particular environmental conditions facilitate (or inhibit) family group co-occurrence. For this purpose, *Cuzick & Edwards*' (*1990*) test was used to assess the null hypothesis of a random geographic distribution of multi-family logs, as this test can detect global spatial clusters in individual-level data. By treating multi-family logs as cases and assessing clustering relative to controls (i.e., single haplotype logs, other multi-haplotype logs, or both classes combined), this method has the advantage of using only those geo-spatial coordinates that were actually sampled. The procedure involved calculating the test statistic $Tk$ (i.e., the number of cases that neighbor other cases, where $k$ is the number of nearest neighbors to consider) from the empirical data, and then comparing $Tk$ to a null distribution generated via Monte Carlo randomizations of case-control labels for each of the spatial locations (9,999 iterations), with significance assessed at the upper tail. Given that the most appropriate neighborhood size is not known *a priori*, iterations of *Cuzick & Edwards*' (*1990*) test was run for values of $k = 1$ to 5, and a Bonferroni correction was used to account for multiple testing. The test

was performed for the following data partitions: multi-family logs ($n = 19$ cases) *vs.* other multi-haplotype logs ($n = 22$ controls), single haplotype logs ($n = 47$ controls), or both classes combined ($n = 69$ controls). All spatial clustering analyses were implemented in CLUSTERSEER v2.5.2 (BioMedware, Ann Arbor, MI, USA). To examine the influence of topography on family group co-occurrence, the null hypothesis of no difference in mean elevation between multi-family logs *vs.* other multi-haplotype logs, single haplotype logs, or both classes combined, was assessed via two-tailed *t*-tests assuming equal variances (as determined using *F*-tests).

## Genetic differentiation among haplotypes

The third goal of this study was to quantify levels of genetic differentiation among different family group matrilines that co-occur within a log, as this would provide insights into how sub-socialilty could impact genetic structure within *C. punctulatus*. By extension, outcomes of this component could point towards potential benefits (or costs) of multi-family living. To quantify levels of genetic divergence among co-occurring haplotypes, MEGA v6.06 (*Tamura et al., 2013*) was used to calculate the number of nucleotide differences between individuals within each multi-family and other multi-haplotype log, and these values were then plotted as a frequency distribution. Within logs, redundant haplotypes were omitted, so that only non-zero nucleotide differences were tallied. In the case of multi-family logs, comparisons that could be clearly identified as representing between-family comparisons (e.g., when two co-occurring females each had a different mtDNA haplotype) were partitioned from those that could not (e.g., when two males and a female each had a different haplotype). To assess whether multi-family *vs.* other multi-haplotype logs exhibited similar distributions of pairwise nucleotide differences, they were compared via a paired two-sample *t*-test.

# RESULTS

## Sampling and rotting log classification

A total of 155 unique mtDNA haplotypes were identified. Of the 88 rotting logs from which *C. punctulatus* were sampled, 47 (53%) contained only a single haplotype, whereas 41 logs had two or more different haplotypes. Of the latter group, 19 logs unambiguously contained multiple families (i.e., 22% overall were multi-family logs), and 22 logs (25% overall) were classified as other multi-haplotype logs (Fig. 1 and Appendix S1). There was a significant female bias in multi-family logs (71% females, $n = 56$ observations, $\chi^2 = 10.286$, $d.f. = 1$, $P = 0.001$). Conversely, there was no meaningful departure from a 1:1 sex ratio in other multi-haplotype logs (54% females, $n = 116$ observations, $\chi^2 = 0.862$, $d.f. = 1$, $P = 0.353$).

## Spatial clustering

*Cuzick & Edwards*' (*1990*) tests showed that there was no clearly detectable departure from the null hypothesis that multi-family logs have a random geographic distribution when the control group was represented by other multi-haplotype logs, or single haplotype logs. Conversely, when single haplotype and other multi-haplotype logs were combined to form

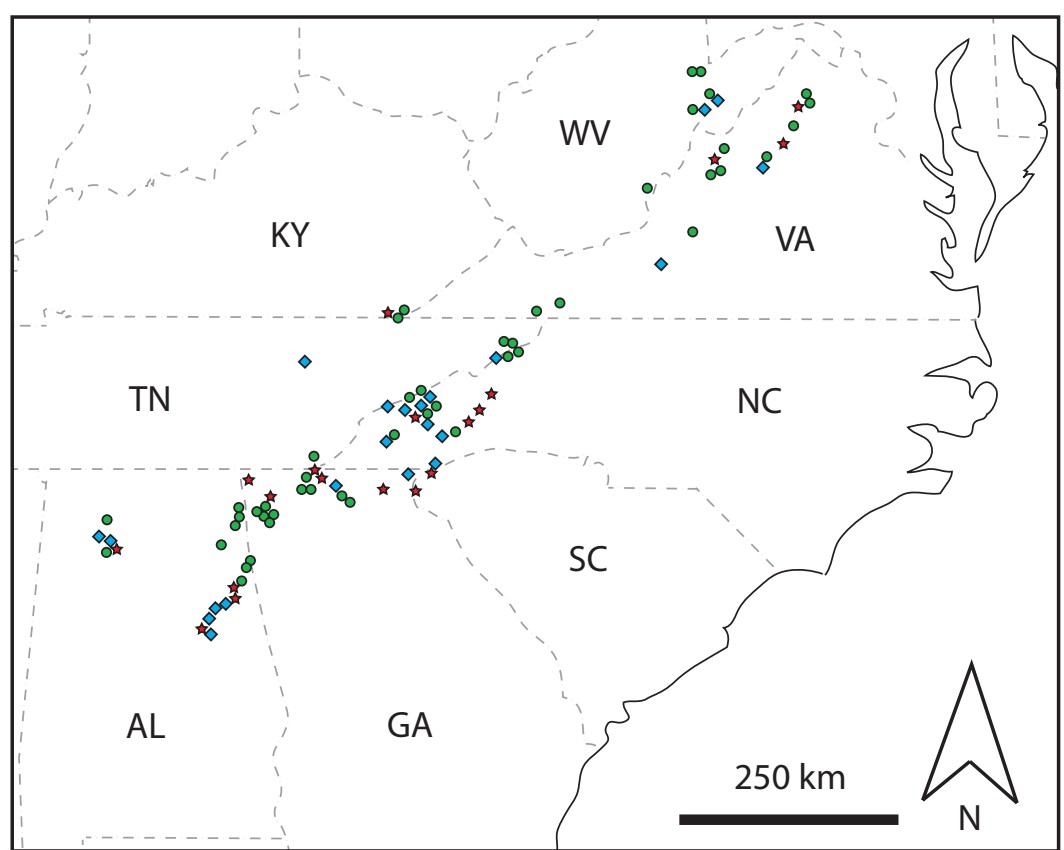

**Figure 1   Map showing spatial distributions of classified rotting logs.** Map of the southeastern USA showing spatial distributions of rotting logs classified as single haplotype (green circle), multi-family (red star), or other multi-haplotype (blue diamond) logs, based on mtDNA sequence data coupled with information on the sex of *C. punctulatus* individuals. State abbreviations are: Alabama, AL; Georgia, GA; Kentucky, KY; North Carolina, NC; South Carolina, SC; Tennessee, TN; Virginia, VA; and West Virginia, WV.

the control group, the test statistic was significant after Bonferroni correction ($P = 0.045$). However, only one iteration of the test generated this result (i.e., when a neighborhood size of $k = 2$ was assumed). All other neighborhood sizes that were examined (i.e., $k = 1, 3, 4$ and $5$) yielded non-significant test statistic values. There was no evidence for elevational partitioning of multi-family *vs.* single haplotype logs, other multi-haplotype logs, or both classes combined ($t = 0.031$, *d.f.* $= 64$, $P = 0.975$; $t = -1.042$, *d.f.* $= 39$, $P = 0.304$; and $t = -0.430$, *d.f.* $= 86$, $P = 0.668$, respectively).

## Genetic differentiation among haplotypes

The sequence alignment of 155 unique mtDNA haplotypes contained 308 polymorphic sites, with a maximum 116 nucleotide differences between a pair of haplotypes (i.e., 10.3% uncorrected sequence divergence). Of the 41 logs with multiple haplotypes, 38 logs contained two different haplotypes, whereas three logs had three different haplotypes. Accordingly, the frequency distribution of nucleotide differences between haplotypes sampled from within the same rotting log was calculated from a total of 47 pairwise comparisons. This frequency distribution showed that while co-occurring haplotypes most
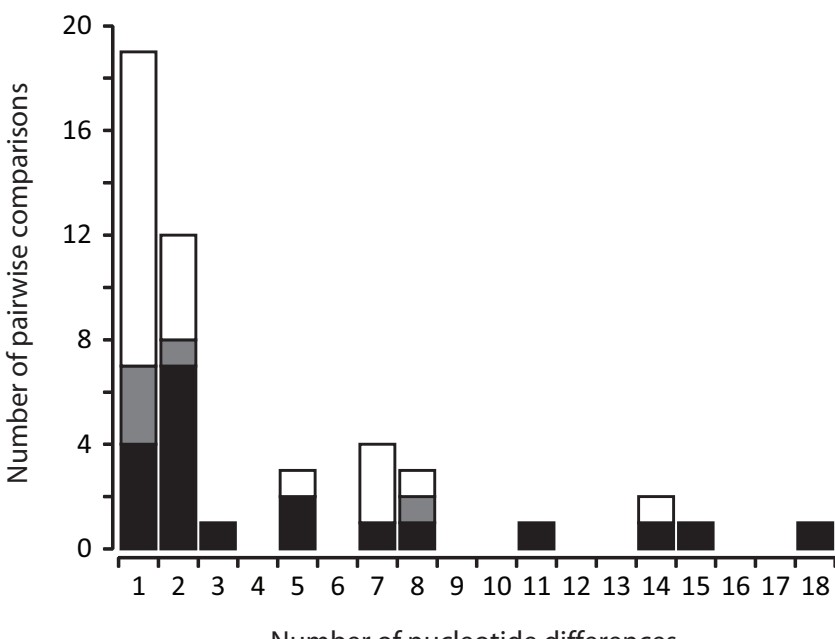

**Figure 2 Frequency distribution of the number of nucleotide differences between pairs of non-redundant mtDNA haplotypes from *C. punctulatus* individuals sampled from the same rotting log.** Shading of bars represents the proportional contribution of multi-family (black plus grey) vs. other multi-haplotype (white) logs to the overall tally. The bi-partitioning of multi-family logs delineates comparisons that could (black) vs. could not (pale grey) be clearly identified as representing between-family comparisons.

often (66% of the time) differ from one another by only one or two mutations, this is not always the case—modest differences within logs (up to 18 mutations, or 1.6% uncorrected sequence divergence) do also occur (Fig. 2). Within multi-family logs, data points that were not clearly attributable to between-family comparisons were relatively few (20%; Fig. 2). Furthermore, there was no strong discord in the distribution of pairwise nucleotide differences between multi-family *vs.* other multi-haplotype logs ($t = 0.410$, *d.f.* = 17, $P = 0.343$).

## DISCUSSION

This paper represents the first explicit investigation of family group co-occurrence in *C. punctulatus*—an evolutionarily important woodroach from a montane forest biodiversity hotspot (*Garrick, 2011*). Broad geographic sampling was coupled with sequencing of multiple individuals per rotting log in order to address the following questions: (1) what proportion of rotting logs contain two or more different mtDNA haplotypes and how often can this be attributed to multiple families inhabiting the same log, (2) are multi-family logs spatially clustered, and (3) what levels of genetic differentiation among haplotypes exist within a log, and how genetically similar are matrilines of co-occurring family groups? The first question provides insights into an as-yet unknown aspect of the basic population biology of *C. punctulatus*, whereas the second question is

exploratory, with the potential to indicate whether environmental factors might influence family group co-occurrence. The third question yields information into how sub-socialilty could impact genetic structure within this species. Together, they provide a framework for subsequent studies. Below, major findings are summarized, and limitations of the present work and recommendations for future studies are also highlighted.

## Family group co-occurrence

Approximately half (47%) of the rotting logs from which *C. punctulatus* were sampled contained multiple mtDNA haplotypes (Fig. 1 and Appendix S1). In contrast, *Kambhampati, Luykx & Nalepa (1996)* sequenced portions of two mtDNA genes (12S and 16S rRNA; 440-bp; and 406- to 414-bp, respectively) for two *C. punctulatus* individuals from each of five sites across Virginia, North Carolina, Georgia and Alabama, but found no within-site genetic differences. However, given that the latter study was limited by small sample sizes, it does not provide a strong comparison here. Similarly, whereas the present study found that almost half of the confirmed multi-haplotype logs (22% overall) clearly contained representatives of two or more family groups (Fig. 1 and Appendix S1), other molecular studies have routinely pooled *C. punctulatus* individuals sampled from different logs (e.g., *Burnside, Smith & Kambhampati, 1999*; *Hossain & Kambhampati, 2001*) and so they too provide no useable comparative data.

One notable exception is a study by *Aldrich, Kambhampati & Krafsur (2005)*, who used allozyme markers to screen genetic variation in ∼40 *C. punctulatus* nymphs from each of 23 sites across the southern Appalachian Mountains. Although samples were often pooled across logs, for 10 sites they were not, and so in these particular cases a single rotting log represented the basic unit of analysis. For this subset of logs, the authors reported that genotype frequencies deviated significantly from Hardy-Weinberg expectations. In nine out of 10 logs the deviation was in the direction of homozygote excess—an outcome that is consistent with Wahlund effect (i.e., genetic substructure, potentially caused by sampling different family groups). However, homozygote excess is also compatible with inbreeding. Notably, recent work by *Yaguchi et al. (in press)* suggested that mating among close relatives is relatively uncommon in *C. punctulatus*. Those authors found that in a sample of 36 mate-pairs genotyped with a set of nuclear microsatellite loci, 72% were unrelated (the remaining 28% had parent–offspring, full-sib, or half-sib relationships). Given this, *Aldrich, Kambhampati & Krafsur*'s (*2005*) data can be explained by frequent sampling of two or more family groups per log, in agreement with conclusions of the present study.

The finding that multi-family logs are quite common in *C. punctulatus* is interesting, as there are reasons why single family logs could be advantageous. For example, inhabitants of single family logs would be released from intraspecific competition, including energetically expensive defense of galleries against intruders (*Nalepa, 2015*). Also, high relatedness among all individuals in a single family log could potentially enhance cooperative behaviors (e.g., dislodgment and reduction of dead wood into small particles that can be ingested; *Watanabe & Tokuda, 2010*), since benefits are received exclusively by kin. However, despite

the potential costs of sharing a log with other families, in *C. punctulatus* this appears to be relatively common.

## Spatial distribution of multi-family logs

Rotting logs that contained representatives of multiple family groups were randomly arrayed across elevational strata, and also across the geographic area that was sampled in this study. Although one of several iterations of *Cuzick & Edwards'* (*1990*) test for spatial clustering did indicate that some structure may exist, this outcome was confined to a narrow portion of the parameter space, and so it is generally poorly supported. Overall, the analyses presented here suggest that if habitat characteristics do promote (or limit) family group co-occurrence, they probably operate over finer scales than those examined in this study. Previously, it has been suggested that large logs may have greater potential to harbor multiple mate-pairs of *C. punctulatus* than small logs (*Yaguchi et al., in press*). Log diameter, length, and other microhabitat characteristics that affect saproxylic invertebrates such as decomposition class and moisture content can certainly vary over fine geographic scales in montane areas (e.g., *Barclay, Ash & Rowell, 2000*; *Woodman, Ash & Rowell, 2006*). Accordingly, an examination of factors that influence family group co-occurrence in *C. punctulatus* should incorporate local environmental variables. As these data are generally not available at the appropriate resolution from remote sensing and other GIS databases, techniques for measuring very fine-scale ecological data would need to be employed (e.g., *Barclay, Ash & Rowell, 2000*; *Grove, 2002*).

## Genetic differentiation

The frequency distribution of nucleotide differences between unique haplotypes sampled from the same rotting log was strongly right-skewed, with most co-occurring haplotypes only one or two mutations apart, but with some occurrences of more substantial divergences (up to 18 mutations, or 1.6% uncorrected sequence divergence; Fig. 2). Also, there was no strong difference between frequency distributions derived from logs that clearly contained multiple families and those that did not (i.e., multi-family *vs.* other multi-haplotype logs). This suggests that most co-occurring families share a recent common ancestor. Furthermore, for multi-haplotype logs that potentially contained only a single family (i.e., where a single mate-pair was sampled), the right-skewed frequency distribution indicates that, based on mtDNA data, parental individuals are not all that distantly related. Although *Yaguchi et al. (in press)* showed that 72% of mate pairs were unrelated (i.e., they shared no more microsatellite alleles per locus than was expected by chance, given the local frequency of each allele), nuclear genotypic data are most informative over relatively short generation-to-generation timescales. Conversely, DNA sequence data can provide insights over deeper time scales (*Sunnucks, 2000*; *Garrick & Sunnucks, 2006*; *Garrick, Caccone & Sunnucks, 2010*; *Garrick et al., 2015*). For instance, mate-pairs that are second cousins would probably not register as being related on the basis of a small set of microsatellite loci due to several generations of gametic recombination, but they could nonetheless be identified as close relatives on the basis of their more slowly evolving mtDNA haplotype. Thus, the notion that co-occurring families share a recent common ancestor is not incompatible with *Yaguchi et al.*'s (*in press*) findings.

Generally speaking, high genetic diversity among individuals that co-occur at a site promotes long-term persistence of the local population (*Frankham, 2005*, and references therein). However, in circumstances where antagonistic and competitive intraspecific interactions mostly occur among non-siblings, high within-site relatedness may be advantageous (*Caesar, Karlsson & Forsman, 2010*). Baseline data from the present study indicate that compared to genetically polymorphic single family logs, there is no meaningful increase in within-site mtDNA diversity among *C. punctulatus* that share their rotting log with other families. Although mtDNA sequence divergence may be only loosely correlated with genome-wide nuclear genetic diversity (*Zhang & Hewitt, 2003*), the empirical data nonetheless suggest that if there are benefits to living in a multi-family log, increased local genetic diversity is unlikely to contribute to this.

In addition to characterizing aspects of family group co-occurrence for the purpose of understanding sub-sociality, the present study also provides an opportunity to examine the adequacy of randomly sampling a single *C. punctulatus* individual per rotting log (e.g., *Steinmiller, Kambhampati & Brock, 2001*; *Aldrich, Krafsur & Kambhampati, 2004*; *Aldrich, Zolnerowich & Kambhampati, 2004*). Based on levels of genetic differentiation among co-occurring haplotypes seen here, the one-sample-per-log strategy would fail to fully capture within-log diversity about half of the time. Even though the majority of co-occurring haplotypes have few mutational differences, some divergences are more substantial (Fig. 2). Indeed, *C. punctulatus* from highly divergent genetic lineages—including those that likely differ in chromosome number—may occasionally co-inhabit the same log (*Garrick, 2016*). Accordingly, depending on the goals of the study, it may be prudent to sample multiple individuals from a log.

## Limitations and future directions

Several limitations of this study warrant consideration. First, while the overall sample size and geographic coverage was large, sample sizes per log were quite small. Accordingly, it was not possible to have a *confirmed absence* of multiple families (or haplotypes) within a given log. This may have impacted the ability to detect spatial clustering, as *Cuzick & Edwards*' (*1990*) test assumes that controls were correctly classified. This also means that the reported frequencies of co-occurrence of multiple haplotypes (47% of logs) and family groups (22% of logs) represent lower bounds, not point estimates. This consideration may partly reconcile the much higher frequency of multi-family logs (90% of relevant logs) suggested by *Aldrich, Kambhampati & Krafsur*'s (*2005*) allozyme data. However, testing of Hardy-Weinberg equilibrium is not an ideal framework for making inferences about family group co-occurrence as there are several possible causes for departures from expected genotype frequencies, and so *Aldrich, Kambhampati & Krafsur*'s (*2005*) data may provide an overestimate. A second limitation of the present study is that the mtDNA-based inference framework has a reduced ability to detect family group co-occurrence in logs from which few females are sampled (e.g., a significantly female-biased sex ratio was detected within multi-family logs, but not within other multi-haplotype logs; also see Table 1). This means that resolution of the approach used here may not be equal across all logs. Third, the simplifying assumption of no *de novo* mutations may not hold true. However,

the mtDNA mutation rate would need to be exceptionally high to overturn this study's repeated inferences of multiple families within a log, and so this limitation is a source of noise (cf. positively misleading).

For understanding family group co-occurrence in *C. punctulatus*, neither a mtDNA-only approach, nor a nuclear marker-only approach, is flawless. The clearest insights would be gained by coupling genotypic data from multiple bi-parentally inherited nuclear markers with haplotypic data from maternally inherited mtDNA, together with information on the sex (and also age cohort) of each individual. Dense sampling—without pooling across logs—is also important, and it would be beneficial to quantify characteristics of the log itself so that potential influences of microhabitat on family group co-occurrence could be investigated. Thus, the present study represents a baseline against which follow-up studies can be compared, and the current findings should be considered a working hypothesis, to be refined with additional data.

## ACKNOWLEDGEMENTS

Sequencing was performed by staff at Yale University's DNA Analysis Facility on Science Hill. RJ Worthington assisted with cockroach sex identifications, RJ Dyer, CQ Hyseni and RE Symula assisted with some fieldwork, and RN Yi contributed to some laboratory work. This paper benefited from constructive comments from DPW Huber and two anonymous reviewers.

### Funding

This work was supported by grants from the American Philosophical Society, Conservation and Research Foundation, National Geographic Society, Bay and Paul Foundations, and start-up funds from the University of Mississippi. The funders had no role in study design, data collection and analysis, decision to publish, or preparation of the manuscript.

### Grant Disclosures

The following grant information was disclosed by the author:
American Philosophical Society.
Conservation and Research Foundation.
National Geographic Society.
Bay and Paul Foundations.
University of Mississippi.

### Competing Interests

The author declares there are no competing interests.

### Author Contributions

- Ryan C. Garrick conceived and designed the experiments, performed the experiments, analyzed the data, contributed reagents/materials/analysis tools, wrote the paper, prepared figures and/or tables, reviewed drafts of the paper.

## Field Study Permissions

The following information was supplied relating to field study approvals (i.e., approving body and any reference numbers):

Scientific collecting permits were issued by the Alabama DCNR, Georgia DNR (29-WBH-12-16), USDA Forest Service, and US National Park Service (GRSM-2012-SCI-2242; SHEN-2012-SCI-0015).

## DNA Deposition

The following information was supplied regarding the deposition of DNA sequences:

GenBank accession numbers: KX944872–KX945357, and KU609620, KU609621, KU609623, KU609624.

## Data Availability

The raw data has been supplied as supplementary files (Appendix S1 and S2).

## Supplemental Information

Supplemental information for this article can be found online at http://dx.doi.org/10.7717/peerj.3127#supplemental-information.

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
