# Peer review of "Genetic insights into family group co-occurrence in Cryptocercus punctulatus, a sub-social woodroach from the southern Appalachian Mountains"

_PeerJ, doi:10.7717/peerj.3127_

## Round 0.1 · original submission · Major Revisions

Thank you to both reviewers for their comments.

This comment from Reviewer 1, I think, summarizes my overall decision as well:

"In short, the main opportunity to improve the paper is to explain the underlying issues, which should determine the methods used, and then to return to those issues in the discussion."

In other words, as pointed out by Reviewer 1, a better presentation of the goals of the work and then writing with those goals in mind would be helpful. The reviewer has provided a good way forward for this.

Reviewer 2 pointed out that there is no mention of voucher specimens. If the author deposited vouchers, please include that in the paper. If not, please do so assuming that the samples are still extant. Physical vouchers plus DNA would be even more helpful. But in either case, vouchering is a substantial consideration here.

Overall the reviewers' opinions were positive, with Reviewer 1 stating that "(t)he evidence is convincing." The changes required, though, are fairly substantial so I am putting this into the major revisions category and so the revised submission (with a point-by-point rebuttal) may go back out for review or re-review.

Reviewer 1 ·

Basic reporting

The paper passes the "basic reporting" criteria.

Experimental design

This is not an experimental study.
The three questions addressed are stated clearly, but the underlying goals are not explained; therefore, it's hard to evaluate the relevance and meaning of the questions. (See below.)

Validity of the findings

As acknowledged in the paper, the methods used will underestimate the proportion of logs that contain multiple families.

Concerning the third question (the level of divergence between haplotypes in the same log), the methods used do not show how similar haplotypes are within logs relative to random expectation, so there is no basis for invoking philopatry or low dispersal (lines 282-284).

The description of skew in the distribution of nucleotide differences is backwards. The distribution has right or positive skew (a long right tail), rather than left skew.

Additional comments

This study used molecular (mtDNA) markers to determine whether rotting logs sometimes support more than one family of the social woodroach Crytocercus punctulatus. The evidence is convincing. The existence of three haplotypes, or two haplotypes among adult females, is inconsistent with the presence of a single family group founded by one male-female pair. Paternal inheritance of mtDNA or rare mutations are not likely to produce those genotype patterns at the observed frequencies.

Although the three questions addressed in this study are stated clearly, not much is said about why they were chosen. It would help to identify the underlying issue(s) and to indicate in the discussion how the results bear on those issues. Why does it matter whether family groups co-occur in the same log? Connections to termite evolution or to events in the Pleistocene (lines 57-62) seem to be weak. Some studies are cited lines 72-73 (also line 298) that have “implicitly assumed that each rotting log is inhabited by a single family.” But the first of those studies considered whether multiple species co-occur in the same geographic area and the other two looked for genetic markers or aspects of the genitalia that differed among species, so it doesn’t appear that they relied on assumptions about how many family groups of C. punctulatus live in one log. (I am not an author of any of those papers.)

Similarly, why investigate whether multi-family logs are clustered? If the goal is to evaluate whether logs with multiple families are more common in some subset of environmental conditions, then there are better approaches such as spatial regressions. The test for spatial clustering is weakened by conducting multiple tests with Bonferroni corrections at different neighborhood sizes, which was done because the most appropriate size is not known a priori (lines 159-160). The most appropriate size depends on the question, so be clear about the goal of the analysis.

The third question concerns the degree of genetic differentiation among haplotypes within logs. Again, why? And why ask whether MF and OMF logs differ in this respect? If the goal is to determine whether co-occurring families tend to be related, then some other approach is needed (because related families sharing the same haplotype will be scored as a single family).

In short, the main opportunity to improve the paper is to explain the underlying issues, which should determine the methods used, and then to return to those issues in the discussion.

Some other suggestions:

I think there’s room in the abstract to say how multi-family logs were detected: the existence of three haplotypes, or two haplotypes among adult females, is inconsistent with the presence of a single family group founded by one male-female pair.

Include more about how the collection was done. Presumably, tracing the gallery systems and looking for barriers (as in Nalepa 1984) would show whether separate groups were present, but would be slow. Was an effort made to collect all adults, or were just the first 2 or 3 taken? This information would help to evaluate the degree to which the % of multi-family logs is underestimated.

My preference would be to see MF spelled out as multi-family throughout the paper.

By analogy with the vocabulary of community ecology, it might be better to say that multiple families “co-occur” rather than “co-exist” within the same log. The latter term implies stability.

lines 189-191: Include the percentages.

line 195: better to say that no departure from the null hypothesis could be detected confidently
See also line 261.

line 199: missing word(s) in “attributable to only [one] iteration of the test”

line 222: missing word in “This paper represents the first explicit investigation [of] family group coexistence...”

line 236: Why “In contrast....”? As noted, the results of the study by Kambhampati et al. can’t be compared to the current study because only two adults were collected, so it would not be possible to detect multi-family logs.

line 299: strike “it” from “it the single sample strategy”

line 330: strike “nor” from “approach, nor, is flawless”

line 332: add “with” to “together [with] information on the sex”

Table 2 legend: Change “are unambiguously comprised of” to “unambiguously contain” or “house.”

Reviewer 2 ·

Basic reporting

Strongly presented.

Experimental design

Sufficient for question asked, limitations are well presented in the discussion.

Validity of the findings

Valid.

Additional comments

The manuscript was well presented. The author placed his work importantly within the literature, and he clearly outlined the work’s merits and caveats.

Given that there is some confusion of the taxonomy (as outlined by the author) Vouchers should have been kept, beyond the sequence data. If that was the case, please add details of location and numbers for vouchers.

The natural history of these animals is not familiar to me, so the sampling of adults was not clear. Were broods present in all cases, and in the three years the brood remains with the parents does the brood reach adulthood? Were all adults sampled, and if not, was there any effort to sample from different parts of the log? Some clarification of sampling should be given in the methods, and some additional detail of natural history could be added to the intro.

Typo in sentence starting on line 272 and ending on line 274. Poor sentence starting on line 289 and ending on 291. Typo in sentence starting on line 298 and ending on line 300. Typo in sentence starting on line 272 and ending on line 274. Extra ‘nor’ on line 330.

---

## Round 0.2 · Major Revisions

I vacillated a bit between major and minor revisions, as the revisions are substantive but do not require any further data collection or analysis. I will wait until I see the response by the authors to the various major (and minor) points and may or may not send it back for one more look by the current reviewer at that point.

Please provide a detailed rebuttal along with a "track changes" and a clean copy of the revised MS as per PeerJ guidelines.

Thank you the the reviewers and the authors for your work and your professionalism during this process.

Reviewer 1 ·

Basic reporting

The paper passes the basic reporting criteria.

Experimental design

It’s not clear to me that the 3rd research question is meaningful. (See the general comments below.)

Validity of the findings

The paper passes the validity criteria.

Additional comments

The manuscript has been improved. The introduction is better at providing context and the enhanced description of collecting methods is very helpful. Genetic studies are needed for tax in which direct observation of breeding and social interactions is difficult.

Some of my comments are similar to those made for the previous version.

My main recommendation was to say more about why the three questions raised in this study were chosen. The presentation of question 1 (how often multiple haplotypes and multiple families occur in the same log) is fine. The paper makes valuable contributions to understanding the biology of woodroaches. The justification for question 2 (whether multiple-family logs are spatially clustered) is implied, but why not state it up front? A single sentence will do the job. Presumably, a non-random pattern would imply that multi-family logs occur at greater frequency in particular conditions (which would remain unknown).

I don’t see any stated justification for question 3 (“what levels of genetic differentiation among haplotypes exist within a log, and how genetically similar are matrilines of co-occurring family groups”). The motivation is not obvious and the discussion of that part of the paper seems misleading because it implies that this part of the study reveals something about the degree of relatedness between families within logs. If co-occurring families were more closely related than other groups within the same local population, that would tell us something about dispersal distances and the formation of new family groups; furthermore, relatedness could affect how groups interact. That seems to be what the author has in mind. For example, it is suggested (323-326) that mtDNA markers are better than microsatellites at detecting second cousins. I think that’s wrong: the mtDNA haplotypes aren’t diverse enough and the number of nucleotide substitutions is irrelevant. Also (lines 318-319), “the right-skewed frequency distribution indicates that parental individuals are not all that distantly related.” This doesn’t tell us about the relatedness that appears in theory about social evolution.

Yes, sequence data can tell us about changes over longer time scales, but it’s not clear to me whether any questions are being asked for which those time scales are relevant. If so, tell the reader more about them. The sentences beginning on lines 316 and 327 say that most co-occurring families share a recent common ancestor and the abstract states that “most haplotypes were only one or two mutations apart, but more substantial divergences ... do occasionally occur within logs.” But that just reflects the diversity of haplotypes in the population.

Greater care is needed in describing limitations of other studies. On lines 70-72, it is implied that studies by previous investigators were “based on the premise that genetic differences among individuals within a log are negligible and so a single random sample is sufficient (e.g., Steinmiller et al. 2001; Aldrich et al. 2004a,b).” But the study by Steinmiller et al. (at least) does not rely on such a premise. The asked whether species’ ranges overlap and whether there are any cryptic species. Their paper seems to be completely agnostic on whether one sample is sufficient to measure genetic differences within logs. The adequacy of sampling one individual per log (lines 341-3) depends on the goals of the study. That method seems to be fine for the goals of Aldrich et al. and Steinmiller et al.

Smaller points:

line 26: “and 19 of these logs” It’s a minimum estimate, so this could say “at least 19”

lines 15-16: “in order to provide baseline information for extending on studies into the evolution ..” The phrase "for extending on studies" doesn’t read right to me. Maybe: “in order to provide information that can be extended by studies on the evolution...”

line 51: “The most well-studied members of Cryptocercus...”
Could be changed to “The best studied members of Cryptocercus...”

286-287: “However, despite the potential costs of sharing a log with other families, baseline data from the present study suggest that costs may be negligible.” I don’t see how this tells us anything about costs. Territorial competition can be common and yet costly.
Here and elsewhere, “baseline” just means basic, rather than a reference point for comparison. I would drop it here.

---

## Round 0.3 · accepted · Accept

While there is still some point of worthy discussion between the author and the reviewer (particularly microsatellites vs mtDNA for determining relatedness in this context), I believe that the authors have provided a good rationale. However, it would be preferable for the author to choose to publish the review history alongside this MS so the future readers can see this (and other) important discussion.

Thanks again to the reviewers and the author for their work on this.